# Sociodemographic inequality in COVID-19 vaccination coverage among elderly adults in England: a national linked data study

Vahe Nafilyan ![ORCID] ,[1,2] Ted Dolby,[1] Cameron Razieh,[3,4,5] Charlotte Hannah Gaughan ![ORCID] ,[6] Jasper Morgan,[1] Daniel Ayoubkhani,[6] Sarah Walker,[7] Kamlesh Khunti,[3,4,5] Myer Glickman,[1] Thomas Yates[3,4,5]

For numbered affiliations see end of article.

**Correspondence to**
Dr Vahe Nafilyan;
vahe.nafilyan@ons.gov.uk

## ABSTRACT

**Objective** To examine inequalities in COVID-19 vaccination rates among elderly adults in England.

**Design** Cohort study.

**Setting** People living in private households and communal establishments in England.

**Participants** 6 655 672 adults aged ≥70 years (mean 78.8 years, 55.2% women) who were alive on 15 March 2021.

**Main outcome measures** Having received the first dose of a vaccine against COVID-19 by 15 March 2021. We calculated vaccination rates and estimated unadjusted and adjusted ORs using logistic regression models.

**Results** By 15 March 2021, 93.2% of people living in England aged 70 years and over had received at least one dose of a COVID-19 vaccine. While vaccination rates differed across all factors considered apart from sex, the greatest disparities were seen between ethnic and religious groups. The lowest rates were in people of black African and black Caribbean ethnic backgrounds, where only 67.2% and 73.8% had received a vaccine, with adjusted odds of not being vaccinated at 5.01 (95% CI 4.86 to 5.16) and 4.85 (4.75 to 4.96) times greater than the white British group. The proportion of individuals self-identifying as Muslim and Buddhist who had received a vaccine was 79.1% and 84.1%, respectively. Older age, greater area deprivation, less advantaged socioeconomic position (proxied by living in a rented home), being disabled and living either alone or in a multigenerational household were also associated with higher odds of not having received the vaccine.

**Conclusion** Research is now urgently needed to understand why disparities exist in these groups and how they can best be addressed through public health policy and community engagement.

## INTRODUCTION

The UK began an ambitious vaccination programme to combat the COVID-19 pandemic on 8 December 2020; by 24 April 2021, 64% of the UK adult population have received their first of the dose.[1]

### Strengths and limitations of this study

► The main strength of our population-level dataset is the availability of a wide range of sociodemographic characteristics not included in electronic health records, allowing for a detailed examination of inequalities in vaccination coverage.

► We presented vaccination rates and Odds Ratios for non-vaccination adjusted for a range of factors to understand further inequalities in vaccination coverage.

► The main limitation is that most demographic and socioeconomic characteristics were derived from the 2011 Census and therefore are 10 years old.

► Because the dataset is based on the 2011 Census, it excluded people living in England in 2011 but not taking part in the 2011 Census, respondents who could not be linked to the 2011–2013 National Health Service Patient Register and recent migrants.

Previous research demonstrates that vaccination rates tend to be lower among certain ethnic groups, and in areas of higher deprivation.[2–4] Existing evidence suggests that COVID-19 vaccination rates differ by level of area deprivation, certain underlying health conditions and ethnicity.[5] Far less is known about how COVID-19 vaccination uptake varies by sociodemographic factors, such as religious affiliation, individual socioeconomic status, living in multigenerational household or disability status, factors disproportionately associated with SARS-CoV-2 infection. Understanding which sociodemographic, economic and cultural factors are associated with low vaccination rates has major implications for designing policies that help maximise the vaccination campaign coverage.

This study investigates inequality in vaccination rates among adults aged ≥70 years in England, using population-level

**Table 1** Variables used in the analyses

| Variable | Coding | Source |
|---|---|---|
| Vaccinated | Received a first dose of a COVID-19 vaccine by 15 March 2020 | NIMS |
| Age | Third-order polynomial | 2011 Census |
| Sex | Female, male | 2011 Census |
| Ethnicity | White British, Bangladeshi, Black African, Black Caribbean, Chinese, Indian, Mixed, Other, Pakistani, White other | 2011 Census |
| Religious affiliation | Christian, Buddhist, Hindu, Jewish, Muslim, no religion, other religion, religion not stated, Sikh | 2011 Census |
| Region | Dummy variables representing region of residence | 2019 NHS Patient Register |
| Rural–urban classification | Urban, rural | 2019 NHS Patient Register |
| Index of Multiple Deprivation | Dummy variables representing quintiles of deprivation | 2019 NHS Patient Register |
| Household tenure | Own, social rented, private rented, other | 2011 Census |
| Level of highest qualification | Degree, A-level or equivalent, GCSE or equivalent, no qualification, other | 2011 Census |
| Disability | Non-disabled, disabled (limited a little), disabled (limited a lot) | 2011 Census |
| Body mass index ($kg/m^2$) | <18.5, 18.5–25, 25–30, ≥30, missing | GPES |
| Chronic kidney disease (CKD) | No CKD, CKD3, CKD4, CKD5 | GPES |
| Learning disability | No learning disability, Down's syndrome, other learning disability | GPES |
| | | |
| Cancer and immunosuppression | Dummies for blood cancer, solid organ transplant, prescribed immunosuppressant medication by GP, prescribed leukotriene or long-acting beta blockers, prescribed regular prednisolone | GPES |
| Other conditions | Diabetes, chronic obstructive pulmonary disease, asthma, rare pulmonary diseases, pulmonary hypertension or pulmonary fibrosis, coronary heart disease, stroke, atrial fibrillation, congestive cardiac failure, venous thromboembolism, peripheral vascular disease, congenital heart disease, dementia, Parkinson's disease, epilepsy, rare neurological conditions, cerebral palsy, severe mental illness (bipolar disorder, schizophrenia, severe depression), osteoporotic fracture, rheumatoid arthritis or systemic lupus erythematosus, cirrhosis of the liver | GPES/HES |

GCSE, General Certificate of Secondary Education; GP, general practitioner; GPES, General Practice Extraction Service; HES, Hospital Episode Statistics; NHS, National Health Service; NIMS, National Immunisation Management System.

administrative records linked to the 2011 Census. This enables examination of a wide range of sociodemographic characteristics, currently lacking in previously published studies, in particular ethnicity, religion, different measures of socioeconomic position and those who report being disabled.

## METHODS
### Study data
We linked vaccination data from the National Immunisation Management System (NIMS) to the Office for National Statistics (ONS) Public Health Data Asset (PHDA) based on National Health Service (NHS) number. The ONS PHDA is a linked dataset combining the 2011 Census, mortality records, the General Practice Extraction Service data for pandemic planning and research and the Hospital Episode Statistics. To obtain NHS numbers for the 2011 Census, we linked the 2011 Census to the 2011–2013 NHS Patient

Registers using deterministic and probabilistic matching, with an overall linkage rate of 94.6%. All subsequent linkages were performed based on NHS numbers.

The study population consisted of people aged ≥70 years, alive on 15 March 2020, who were residents in England, registered with a general practitioner and enumerated at the 2011 Census. Of 6 605 315 adults aged ≥70 years who received a first dose of a COVID-19 vaccine in NIMS, 6 242 384 (94.5%) were linked to the ONS PHDA.

### Outcome
The main outcome was having received at least a first dose of a COVID-19 vaccine by 15 March 2021, as recorded in the NIMS data available on 31 March 2021. Phase 1 of the vaccination policy for England aimed to offer a first vaccination appointment to all those ≥70 years by 15 February, and we allowed a further month to ensure full coverage.

| Table 2 | Characteristics of the study population | |
|---|---|---|
| **Variable** | **Level** | **Count (%)** |
| Vaccinated | | 6 202 780 (93.2) |
| Sex | Female | 3 672 314 (55.2) |
| | Male | 2 983 358 (44.8) |
| Age | Mean (SD) | 78.8 (6.5) |
| Ethnicity | Bangladeshi | 11 522 (0.2) |
| | Black African | 21 535 (0.3) |
| | Black Caribbean | 52 883 (0.78) |
| | Chinese | 18 452 (0.38) |
| | Indian | 103 564 (1.6) |
| | Mixed | 24 637 (0.34) |
| | Other | 65 241 (1.0) |
| | Pakistani | 39 723 (0.6) |
| | White British | 6 095 276 (91.6) |
| | White other | 222 839 (3.4) |
| Religion | Buddhist | 16 403 (0.3) |
| | Christian | 5 221 392 (78.5) |
| | Hindu | 61 634 (0.9) |
| | Jewish | 39 800 (0.6) |
| | Muslim | 86 841 (1.3) |
| | No religion | 725 695 (10.9) |
| | Other religion | 22 327 (0.3) |
| | Religion not stated | 449 781 (6.8) |
| | Sikh | 31 799 (0.5) |
| IMD quintile | 1 (most deprived) | 913 809 (13.7) |
| | 2 | 1 140 651 (17.1) |
| | 3 | 1 407 155 (21.1) |
| | 4 | 1 560 023 (23.4) |
| | 5 (least deprived) | 1 634 034 (24.6) |
| Household tenure | Owned | 5 488 126 (82.5) |
| | Private rented | 273 707 (4.1) |
| | Social rented | 778 867 (11.7) |
| | Other (eg, live rent free) | 114 972 (1.7) |
| Rural–urban | Rural | 6 005 144 (82.4) |
| | Urban | 304 412 (4.2) |
| Household composition | 2 elderly | 847 508 (11.6) |
| | 1 elderly | 128 083 (1.8) |
| | Care home | 436 211 (6.0.) |
| | Missing household | 9035 (0.1) |
| | Multigenerational | 620 167 (8.5) |
| | Other (3+ adults) | 714 211 (9.8) |

Adults aged 70 years or over, living in England, alive on 15 March 2021.
IMD, Index of Multiple Deprivation.

## Exposures and covariates

This dataset combines comprehensive sociodemographic information from the 2011 Census with a detailed medical history from clinical records. All individual-level sociodemographic characteristics (ethnic group, religious affiliation, disability status, educational attainment) came from the 2011 Census. We used a 10-category ethnic group classification (White British, Bangladeshi, Black African, Black Caribbean, Chinese, Indian, Mixed, Other, Pakistani, White other). Self-reported religious group, place of residence (region within England, private or care home) and area-based deprivation (Index of Multiple Deprivation[6]) were derived based on the 2019 Patient Register. Comorbidities were defined as in the QCovid risk prediction model, a model used to assess the risk of severe COVID-19 outcomes in the general population, used to inform the prioritisation of the vaccination campaign.[7] All variables included in this analysis are listed in table 1.

## Statistical analyses

First, we estimated the first dose vaccination rates by a range of demographic and socioeconomic characteristics. Second, to understand the drivers of the observed differences in vaccination rates, we used logistic regression to estimate the odds of not having received a first dose of a COVID-19 vaccine. For each exposure, we compared ORs from models adjusted for different sets of covariates. We estimated unadjusted ORs, ORs adjusted for sex and age, and ORs adjusted for all geographical and sociodemographic characteristics, disability status and pre-existing conditions. All analyses were conducted using R V.3.5

## Patient and public involvement

No patient involved.

## RESULTS

Our study population included 6 655 672 adults aged ≥70 years who lived in England. A total of 55.2% were women and the mean age was 78.8 (SD: 6.5) years; 91.6% identified themselves as White British, 78.5% as Christian. A total of 82.5% owned their home (table 2). By 15 March 2021, 93.2% of people living in England aged 70 years and over had received at least one dose of a COVID-19 vaccine.

Table 3 shows vaccination rates by demographic and socioeconomic factors, as well as ORs from different models. Vaccination rates differed across all factors considered, apart from sex. The lowest rates were in people of Black African and Black Caribbean ethnic backgrounds where only 67.2% and 73.9% had received a vaccine. Adjusting for differences in geography, sociodemographic factors and underlying health conditions did not fully explain the lower probability of having received the vaccine among ethnic minority groups. Compared with people of white British ethnicity, the fully adjusted OR for Black African individuals was 5.01 (95% CI 4.86 to 5.16), while the unadjusted OR was 7.62 (7.40 to 7.84),

**Table 3** Vaccination rates and ORs for not being vaccinated by sociodemographic characteristics

| Exposure | Group | Vaccination rate | OR (model 1) | OR (model 2) | OR (model 3) |
|---|---|---|---|---|---|
| Age group | 70–74 | 90.9 (90.8 to 90.9) | 1 (ref) | 1 (ref) | 1 (ref) |
| | 75–79 | 93.8 (93.8 to 93.9) | 0.65 (0.65 to 0.66) | 0.65 (0.65 to 0.66) | 0.66 (0.65 to 0.66) |
| | 80–84 | 95.6 (95.5 to 95.6) | 0.46 (0.46 to 0.46) | 0.46 (0.46 to 0.46) | 0.45 (0.45 to 0.46) |
| | 85–89 | 95.1 (95.0 to 95.1) | 0.51 (0.51 to 0.52) | 0.51 (0.51 to 0.52) | 0.51 (0.51 to 0.52) |
| | 90–94 | 94.0 (93.9 to 94.1) | 0.63 (0.62 to 0.64) | 0.63 (0.62 to 0.64) | 0.64 (0.63 to 0.65) |
| | 95–99 | 92.3 (92.1 to 92.5) | 0.83 (0.81 to 0.85) | 0.82 (0.80 to 0.85) | 0.82 (0.80 to 0.84) |
| | 100+ | 85.5 (84.8 to 86.1) | 1.69 (1.61 to 1.78) | 1.68 (1.60 to 1.77) | 1.60 (1.52 to 1.68) |
| Sex | Female | 93.2 (93.2 to 93.2) | 1 (ref) | 1 (ref) | 1 (ref) |
| | Male | 93.2 (93.2 to 93.2) | 1.00 (1.00 to 1.01) | 0.98 (0.98 to 0.99) | 1.03 (1.02 to 1.03) |
| Disability | Not limited | 93.5 (93.5 to 93.5) | 1 (ref) | 1 (ref) | 1 (ref) |
| | Limited a little | 93.4 (93.3 to 93.4) | 1.02 (1.01 to 1.03) | 1.12 (1.11 to 1.13) | 1.08 (1.07 to 1.08) |
| | Limited a lot | 91.3 (91.2 to 91.3) | 1.37 (1.36 to 1.38) | 1.47 (1.46 to 1.48) | 1.29 (1.28 to 1.30) |
| Ethnicity | White British | 94.0 (94.0 to 94.0) | 1 (ref) | 1 (ref) | 1 (ref) |
| | Bangladeshi | 82.7 (82.0 to 83.4) | 3.26 (3.11 to 3.42) | 3.48 (3.31 to 3.65) | 2.56 (2.43 to 2.69) |
| | Black African | 67.2 (66.6 to 67.8) | 7.62 (7.40 to 7.84) | 7.59 (7.37 to 7.81) | 5.01 (4.86 to 5.16) |
| | Black Caribbean | 73.8 (73.4 to 74.2) | 5.55 (5.44 to 5.66) | 6.35 (6.22 to 6.47) | 4.85 (4.75 to 4.96) |
| | Chinese | 82.8 (82.3 to 83.4) | 3.23 (3.11 to 3.36) | 3.11 (2.99 to 3.23) | 2.64 (2.54 to 2.75) |
| | Indian | 90.9 (90.7 to 91.0) | 1.57 (1.54 to 1.60) | 1.55 (1.52 to 1.59) | 1.35 (1.32 to 1.38) |
| | Mixed | 85.3 (84.9 to 85.7) | 2.69 (2.60 to 2.79) | 2.67 (2.58 to 2.77) | 2.21 (2.14 to 2.30) |
| | Other | 82.9 (82.6 to 83.2) | 3.22 (3.15 to 3.29) | 3.12 (3.05 to 3.18) | 2.44 (2.39 to 2.50) |
| | Pakistani | 79.6 (79.2 to 80.0) | 3.99 (3.89 to 4.09) | 4.12 (4.02 to 4.22) | 3.59 (3.50 to 3.68) |
| | White other | 87.7 (87.5 to 87.8) | 2.20 (2.17 to 2.23) | 2.24 (2.21 to 2.27) | 1.93 (1.90 to 1.95) |
| IMD quintile | 1 (most deprived) | 90.6 (90.6 to 90.7) | 1.77 (1.76 to 1.79) | 1.76 (1.75 to 1.78) | 1.60 (1.59 to 1.62) |
| | 2 | 92.1 (92.0 to 92.1) | 1.48 (1.46 to 1.49) | 1.47 (1.46 to 1.48) | 1.34 (1.33 to 1.35) |
| | 3 | 93.4 (93.4 to 93.5) | 1.20 (1.19 to 1.22) | 1.20 (1.19 to 1.21) | 1.17 (1.16 to 1.18) |
| | 4 | 94.0 (94.0 to 94.0) | 1.09 (1.08 to 1.10) | 1.09 (1.08 to 1.10) | 1.09 (1.08 to 1.10) |
| | 5 (least deprived) | 94.5 (94.4 to 94.5) | 1 (ref) | 1 (ref) | 1 (ref) |
| Religion | Christian | 93.8 (93.8 to 93.9) | 1 (ref) | 1 (ref) | 1 (ref) |
| | Buddhist | 84.1 (83.5 to 84.7) | 2.88 (2.76 to 3.01) | 2.63 (2.52 to 2.74) | 2.03 (1.95 to 2.12) |
| | Hindu | 91.5 (91.2 to 91.7) | 1.43 (1.39 to 1.47) | 1.39 (1.35 to 1.43) | 1.03 (1.00 to 1.06) |
| | Jewish | 93.1 (92.8 to 93.3) | 1.13 (1.09 to 1.18) | 1.12 (1.08 to 1.17) | 0.94 (0.90 to 0.97) |
| | Muslim | 79.1 (78.9 to 79.4) | 4.02 (3.95 to 4.09) | 4.04 (3.97 to 4.11) | 2.74 (2.69 to 2.79) |
| | No religion | 91.9 (91.9 to 92.0) | 1.34 (1.33 to 1.35) | 1.25 (1.23 to 1.26) | 1.23 (1.22 to 1.24) |
| | Other religion | 85.4 (84.9 to 85.8) | 2.61 (2.51 to 2.71) | 2.41 (2.32 to 2.50) | 2.15 (2.07 to 2.23) |
| | Religion not stated | 91.5 (91.4 to 91.6) | 1.42 (1.41 to 1.44) | 1.40 (1.38 to 1.41) | 1.35 (1.33 to 1.36) |
| | Sikh | 91.6 (91.3 to 91.9) | 1.39 (1.34 to 1.45) | 1.35 (1.30 to 1.41) | 1.07 (1.03 to 1.11) |
| Household tenure | Owned | 94.0 (93.9 to 94.0) | 1 (ref) | 1 (ref) | 1 (ref) |
| | Other | 91.1 (90.9 to 91.2) | 1.52 (1.49 to 1.55) | 1.55 (1.52 to 1.58) | 1.49 (1.46 to 1.52) |
| | Private rented | 88.4 (88.2 to 88.5) | 2.05 (2.02 to 2.07) | 1.96 (1.94 to 1.99) | 1.81 (1.79 to 1.83) |
| | Social rented | 89.9 (89.8 to 89.9) | 1.75 (1.73 to 1.76) | 1.77 (1.76 to 1.79) | 1.60 (1.59 to 1.61) |
| Rural–urban | Rural | 94.5 (94.4 to 94.5) | 1 (ref) | 1 (ref) | 1 (ref) |
| | Urban | 92.8 (92.8 to 92.8) | 1.33 (1.32 to 1.34) | 1.34 (1.33 to 1.35) | 1.12 (1.12 to 1.13) |

Continued

**Table 3** Continued

| Exposure | Group | Vaccination rate | OR (model 1) | OR (model 2) | OR (model 3) |
|---|---|---|---|---|---|
| Household composition | 2 elderly | 94.5 (94.4 to 94.5) | 1 (ref) | 1 (ref) | 1 (ref) |
| | 1 elderly | 92.5 (92.5 to 92.6) | 1.38 (1.37 to 1.39) | 1.53 (1.52 to 1.54) | 1.32 (1.31 to 1.33) |
| | Care home | 94.9 (94.8 to 95.0) | 0.92 (0.89 to 0.95) | 1.10 (1.06 to 1.13) | 0.89 (0.86 to 0.91) |
| | Multigenerational | 90.3 (90.3 to 90.4) | 1.83 (1.82 to 1.85) | 1.77 (1.76 to 1.79) | 1.39 (1.38 to 1.40) |
| | Other (3+ adults) | 90.0 (89.6 to 90.3) | 1.91 (1.84 to 1.97) | 1.81 (1.75 to 1.88) | 1.54 (1.49 to 1.60) |

Adults aged 70 years or over, living in England, alive on 15 March 2021. Model 1: unadjusted; model 2: adjusted for sex and age (cubic splines); model 3: adjusted for sex, age (cubic splines), care home status, rural/urban, region, ethnicity (except when looking at religion as an exposure), IMD quintile (except when looking at household tenure as an exposure), disability, BMI and comorbidities. See table 1 for more details on the variables included in the models.
BMI, body mass index; IMD, Index of Multiple Deprivation.

suggesting that geography, sociodemographic factors and pre-pandemic health only explain about 40% of the elevated odds of not being vaccinated.

Vaccination rates also varied markedly across religious groups. While 93.8% of Christians had been vaccinated, only 79.1% of Muslims and 84.1% of Buddhists had been vaccinated. Stark differences remained after adjustment for other factors, with an adjusted OR of not being vaccinated of 2.74 (2.69 to 2.79) for Muslims and 2.03 (1.95 to 2.12) for Buddhists, compared with Christians.

Greater area deprivation, less advantaged socioeconomic position (proxied by living in a rented home), being disabled and living either alone or in a multigenerational household were also associated with low vaccination rates, even when adjusting for other factors (table 3). These differences were less pronounced than the differences between ethnic groups or religious affiliations.

## DISCUSSION
### Main findings
Our analysis using whole population-level linked data in England suggests that first dose vaccination rates in adults aged ≥70 years differed markedly by ethnic group and self-reported religious affiliation. The percentage of people vaccinated was lower among all minority ethnic groups compared with the white British population, with the lowest vaccination rates observed among Black African, Black Caribbean, Bangladeshi and Pakistani individuals. In addition, lower vaccination rates were reported among individuals who identified as Muslim and Buddhist. While some differences were found by deprivation, household factors, disability status and other sociodemographic factors, these were less pronounced compared with ethnicity or religious affiliation.

### Comparison with other studies
Few studies have investigated how COVID-19 vaccination coverage varies by a wide range of sociodemographic characteristics. Our results on ethnicity and area deprivation are consistent with one previous study based on clinical records for 40% of patients in England.[3] In addition, our results confirm studies showing that influenza,

shingles and pneumococcal vaccination are patterned by similar factors, including ethnicity, deprivation and household size.[8] Pre-pandemic, religion and culture have been postulated to be important factors in determining vaccination uptake[9]; our results extend this by showing that self-reported religious affiliation is an important factor in COVID-19 vaccine uptake. Differences in vaccination rate and potential vaccination hesitancy between religious groups may not be based on religious beliefs, but rather reflect safety and other concerns,[10] or, given high infection rates in some of these groups,[11] beliefs that vaccination is not needed after natural infection. We also find that vaccination rates vary by individual characteristics not reported in previous studies, such as household tenure (a proxy for socioeconomic status), household composition and disability status.

### Strengths and limitation
The primary study strength is using nationwide linked population-level data from clinical records and the 2011 Census. Unlike studies based solely on electronic health records, we examined a wide range of sociodemographic characteristics. Unlike surveys, we can precisely estimate vaccination rates and ORs for small groups. The main limitation is that most demographic and socioeconomic characteristics are derived from the 2011 Census and therefore are 10 years old. However, we focus primarily on characteristics that are unlikely to change over time, such as ethnicity or religion, or likely to be stable for our population (adults aged ≥70 years), such as household tenure. However, for the characteristics likely to change over time, such as disability status, the time difference may introduce some bias into the estimates, although this would be expected to dilute differences, since we are most likely missing some long-term health conditions. Care home residency and area deprivation were derived from the 2019 Patient Register and are therefore not subject to the same biases. Another limitation is that because the PHDA was based on the 2011 Census, it excluded people living in England in 2011 but not taking part in the 2011 Census; respondents who could not be linked to the 2011–2013 NHS Patient Register and recent

migrants. Consequently, we excluded 5.4% of vaccinated people who could not be linked to the ONS PHDA.

## CONCLUSION

There are stark differences in COVID-19 vaccination rates by ethnic group and religious affiliation. Research is now urgently needed to understand why these disparities exist in these groups and how they can best be addressed through public health policy and community engagement. Understanding barriers and supporting participation in the vaccine programme is especially important because the groups with low vaccination coverage were also at elevated risk of COVID-19 mortality in the first two waves of the pandemic,[11–14] are associated with factors, such as frailty, that will continue to elevate risk as the pandemic evolves.[15]

**Author affiliations**
[1]Health Analysis and Life Event, Office for National Statistics, Newport, UK
[2]Faculty of Public Health, Environment and Society, London School of Hygiene & Tropical Medicine, London, UK
[3]Diabetes Research Centre, University of Leicester, Leicester, UK
[4]NIHR Leicester Biomedical Research Centre, University of Leicester, Leceister, UK
[5]University Hospitals of Leicester NHS Trust, Leicester, UK
[6]Methodology Division, Office for National Statistics, Newport, UK
[7]Nuffield Department of Medicine, University of Oxford, Oxford, UK

**Acknowledgements** We are grateful to Charlotte Bermingham for useful discussions about this paper.

**Contributors** Study conceptualisation was led by VN, TY and CR. VN, TY, CR, TD, CHG, SW, DA, MG and KK contributed to the development of the research question, study design, with development of statistical aspects led by VN and TD. VN, JM and TD were involved in data specification, curation and collection. VN and TD conducted and checked the statistical analyses. VN, TY, CR, TD, CHG, JM, SW, DA, MG and KK contributed to the interpretation of the results. VN, TY and CR wrote the first draft of the paper. VN, TY, CR, TD, CHG, JM, SW, DA, MG and KK contributed to the critical revision of the manuscript for important intellectual content and approved the final version of the manuscript. VN had full access to all data in the study and takes responsibility for the integrity of the data and the accuracy of the data analysis. The lead author (VN) affirms that the manuscript is an honest, accurate, and transparent account of the study being reported; that no important aspects of the study have been omitted; and that any discrepancies from the study as planned have been explained.

**Funding** This work was supported by a grant from the UKRI (MRC)-DHSC (NIHR) COVID-19 Rapid Response Rolling Call (MR/V020536/1) and from HDR-UK (HDRUK2020.138). KK, TY and CR are supported by the National Institute for Health Research (NIHR) Applied Research Collaboration East Midlands (ARC EM) and the NIHR Leicester Biomedical Research Centre (BRC). SW is an NIHR Senior Investigator and is supported by the NIHR Health Protection Research Unit in Healthcare Associated Infections and Antimicrobial Resistance at Oxford University in partnership with Public Health England (PHE) (NIHR200915) and the NIHR BRC, Oxford.

**Disclaimer** The views expressed in this publication are those of the authors and not necessarily those of the ONS, the NHS, the NIHR, the Department of Health or PHE.

**Competing interests** KK is Director of the University of Leicester Centre for Black Minority Ethnic Health, Trustee of the South Asian Health Foundation, Chair of the Ethnicity Subgroup of SAGE and Member of Independent SAGE.

**Patient and public involvement** Patients and/or the public were not involved in the design, or conduct, or reporting, or dissemination plans of this research.

**Patient consent for publication** Not required.

**Ethics approval** Ethical approval was obtained from the National Statistician's Data Ethics Advisory Committee (NSDEC(20)12).

**Provenance and peer review** Not commissioned; externally peer reviewed.

**Data availability statement** Data are not yet available, but will be made available on the ONS Secure Research Service for accredited researchers.

**ORCID iDs**
Vahe Nafilyan http://orcid.org/0000-0003-0160-217X
Charlotte Hannah Gaughan http://orcid.org/0000-0002-3349-3062

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
