## [Reviewer comments · BMJ Open]

ARTICLE DETAILS

TITLE (PROVISIONAL)	Sociodemographic inequality in COVID-19 vaccination coverage amongst elderly adults in England: a national linked data study
AUTHORS	Nafilyan, Vahe; Dolby, Ted; Razieh, Cameron; Gaughan, Charlotte; Morgan, Jasper; Ayoubkhani, Daniel; Walker, Sarah; Khunti, Kamlesh; Glickman, Myer; Yates, Thomas

VERSION 1 – REVIEW

REVIEWER	Palermo, Sara University of Turin
REVIEW RETURNED	09-Jun-2021

GENERAL COMMENTS	One of the most shocking aspects of the COVID-19 pandemic is how lethal this condition is for the older population. The risk for death and severe illness with COVID-19 is best predicted by age. The likelihood of death increases exponentially with age among those who contract the virus in all countries where this has been examined. The risk of severe illness from COVID-19 increases with age. This is why CDC recommends that adults 65 years and older receive COVID-19 vaccines. Getting a COVID-19 vaccine is an important step to help prevent getting sick from COVID-19. It is essential to emphasise the importance of the Authors' proposed topic for public health policy at the global level. In a cohort study, the Authors considered that first dose vaccination rates in adults aged 70 or over differed markedly by ethnic group and self-reported religious affiliation, even after adjusting for geography, socio-demographic factors and underlying health conditions. Their approach found relevant differences in vaccination rates by deprivation, household composition, and disability status. A point of strength is the choice to consider those sociocultural variables that all too often are not included in the evaluation of possible mediating-moderating factors of treatments (extrinsic factors). A second strong point is represented by samples size (6,829,643 adults aged) which, considering the number of variables analysed, guarantees the robustness of the results and a good possibility of generalising them. For what is my concern the topic is relevant and interesting. The manuscript is well-written and the results clearly presented. I suggest some minor revisions of the manuscript and a small number of majors before its acceptance in order to improve the interpretation of the findings by readers. I remain available for a second evaluation of the manuscript. • The introduction is appreciable. Although, it could contain some more details about the neural correlates of depression and associated pathophysiology. The relationship between disability, social isolation, and treatment compliance. In particular, the
--

	construct of “physical, psychological, and social frailty” conforms well to the model proposed by the Authors. Correctly, the authors discuss "underlying health conditions", "disability status" "living alone" "socio-economic status" and so on. These aspects influence the immunobiography of the elderly. Therefore, I propose an in-depth discussion in the introduction about the impact of frailty on both the risk of acquiring COVID-19 infection and the efficacy of vaccines. For an initial study, I recommend https://doi.org/10.3389/fmed.2020.558835 Starting with articles such as this one, Authors might identify some interesting insights to further their own discussion as well  • With respect to methods, I recommend further detailing the QCOVID risk prediction model. I believe that it could be of great utility for the readers, especially if you are not experts in the field. • The Authors propose a comparison between their results and the findings of other studies. Would it be possible to propose a comparison between countries as well? Are there data indicating how the variables identified influence vaccinations - not only in the UK (representing the advanced western world) - but in Asian, African or South American states? • The Authors conclude that “the groups with low vaccination coverage were also at elevated risk of COVID-19 mortality in the first two waves of the pandemic”. This assertion seems highly plausible. Nevertheless - as I pointed out at the beginning - and influence of ageing processes and frailty seems to be an element that comes into interaction with those identified and influences outcomes. I would recommend an integration on this if it were possible I thank the Authors in advance for their understanding and valuable collaboration in making the proposed changes.
--	---

REVIEWER	McCaffery, Kirsten The University of Sydney, Sydney Health Literacy Lab, School of Public Health
REVIEW RETURNED	12-Jun-2021

GENERAL COMMENTS	This is an excellent and important paper. I only have a few minor suggestions for the authors:  1. Abstract: the conclusion here is out of step with the findings and paper. We do not know if the difference in vaccination rates observed is due to 'hesitancy' - it would be more accurate to reflect the conclusion from the main paper's discussion in the abstract, ie. the results show important disparities by sociodemographic variables including measures of social disadvantage but most strikingly by ethnicity and religion. Policy interventions are now needed to ensure inequalities in COVID-19 outcomes are not even further exacerbated. 2. Minor typos on page 4, line 21, line 26. 3. Methods: could the authors add a statistical analysis section which outlines their methods, models and statistical analysis software. I would prefer to see the supplementary table 1 with the variables used in the analysis in the main paper. I found this useful to fully comprehend the different measures, their source and time of assessment. 4. Table 1 - I am not sure of the convention when you have a sample size this large but I found the use of 2 decimal places for the percentages difficult to read. It would be easier for readers to use only 1 decimal here. Consider a little additional line space
---

	between new variables and less line space between categories within the same variable. It can be hard to ensure you are reading the right line of results. Define IMD in a table footnote. Give an example of 'other' in the household tenure category. Note other is usually the last category of a subset. Table 2 is also not easy on the eye! This might be helped by adding larger line spaces between new variables and reducing the line space with categories within the same variable. 5. Discussion - I think the results are perhaps a little understated. The fact that the odds of being unvaccinated for black african and black caribbean (when other sociodemographic factors are controlled) is still around 5 is pretty shocking. I know that it is important not to over state results but these results are very strong and clear. I think these are very important and there is need to draw attention to the fact that more support is needed to understand barriers in these groups and support participation in the vaccine program for now and in the future.
--	---

VERSION 1 – AUTHOR RESPONSE

Reviewer: 1

Dr. Sara Palermo, University of Turin

Comments to the Author:

One of the most shocking aspects of the COVID-19 pandemic is how lethal this condition is for the older population. The risk for death and severe illness with COVID-19 is best predicted by age. The likelihood of death increases exponentially with age among those who contract the virus in all countries where this has been examined. The risk of severe illness from COVID-19 increases with age. This is why CDC recommends that adults 65 years and older receive COVID-19 vaccines. Getting a COVID-19 vaccine is an important step to help prevent getting sick from COVID-19. It is essential to emphasise the importance of the Authors' proposed topic for public health policy at the global level.

In a cohort study, the Authors considered that first dose vaccination rates in adults aged 70 or over differed markedly by ethnic group and self-reported religious affiliation, even after adjusting for geography, socio-demographic factors and underlying health conditions. Their approach found relevant differences in vaccination rates by deprivation, household composition, and disability status. A point of strength is the choice to consider those sociocultural variables that all too often are not included in the evaluation of possible mediating-moderating factors of treatments (extrinsic factors). A second strong point is represented by samples size (6,829,643 adults aged) which, considering the number of variables analysed, guarantees the robustness of the results and a good possibility of generalising them.

For what is my concern the topic is relevant and interesting. The manuscript is well-written and the results clearly presented. I suggest some minor revisions of the manuscript and a small number of majors before its acceptance in order to improve the interpretation of the findings by readers. I remain available for a second evaluation of the manuscript.

Thank you very much for your comments and suggestions which contributed to improve the paper

- The introduction is appreciable. Although, it could contain some more details about the neural correlates of depression and associated pathophysiology. The relationship between disability, social isolation, and treatment compliance. In particular, the construct of "physical, psychological, and social frailty" conforms well to the model proposed by the Authors. Correctly, the authors discuss "underlying health conditions", "disability status" "living alone" "socio-economic status" and so on. These aspects influence the immunobiography of the elderly. Therefore, I propose an in-depth discussion in the

Thank you very much for the suggestions, which really helped make the paper clearer

1. Abstract: the conclusion here is out of step with the findings and paper. We do not know if the difference in vaccination rates observed is due to 'hesitancy' - it would be more accurate to reflect the conclusion from the main paper's discussion in the abstract, ie. the results show important disparities by sociodemographic variables including measures of social disadvantage but most strikingly by ethnicity and religion. Policy interventions are now needed to ensure inequalities in COVID-19 outcomes are not even further exacerbated.

Response: Thank you for the suggestion, we have redrafted the conclusion:

'Research is now urgently needed to understand why disparities exist in these groups and how they can best be addressed through public health policy and community engagement.'

2. Minor typos on page 4, line 21, line 26.

3. Methods: could the authors add a statistical analysis section which outlines their methods, models and statistical analysis software. I would prefer to see the supplementary table 1 with the variables used in the analysis in the main paper. I found this useful to fully comprehend the different measures, their source and time of assessment.

Response: Thank you for these suggestions. We have created a statistical analyses subsection, where we outline the method and models as well as the statistical software we used. We have also moved supplementary table 1 into the main text

4. Table 1 - I am not sure of the convention when you have a sample size this large but I found the use of 2 decimal places for the percentages difficult to read. It would be easier for readers to use only 1 decimal here. Consider a little additional line space between new variables and less line space between categories within the same variable. It can be hard to ensure you are reading the right line of results. Define IMD in a table footnote. Give an example of 'other' in the household tenure category. Note other is usually the last category of a subset.

Table 2 is also not easy on the eye! This might be helped by adding larger line spaces between new variables and reducing the line space with categories within the same variable.

Response: Thank you for these suggestions which improved the readability of the Tables> For Table 1 (now Table 2), we have rounded the % to one decimal, and moved the other category to the end (and gave an example) We have also added lines to separate the different variables, in both table 2 and table 3

5. Discussion - I think the results are perhaps a little understated. The fact that the odds of being unvaccinated for black african and black caribbean (when other sociodemographic factors are controlled) is still around 5 is pretty shocking. I know that it is important not to over state results but there results are very strong and clear. I think these are very important and there is need to draw attention to the fact that more support is needed to understand barriers in these groups and support participation in the vaccine program for now and in the future.

Response: Thank you for this comment. We have redrafted the conclusion to emphasise our results the importance of our results, and the need for a policy response.

Thank you again for your very useful comments.

VERSION 2 – REVIEW

REVIEWER	Palermo, Sara University of Turin
REVIEW RETURNED	06-Jul-2021

GENERAL COMMENTS	I had the opportunity to read the new version of the manuscript, also taking note of the appropriate and correct evaluations carried out by the other Reviewer who has well integrated with his/her skills the evaluation of the analysis part. Considering my previous remarks, the Authors gave adequate answers and made the proposed integrations within the limits allowed by the editorial guidelines. Regarding the only part of my competence, I consider the manuscript publishable in its current form.
--

VERSION 2 – AUTHOR RESPONSE

- The main strength of our population-level dataset is the availability of a wide range of socio-demographic characteristics not included in electronic health records, allowing for a detailed examination of inequalities in vaccination coverage.
- We presented vaccination rates and odds ratios for non-vaccination adjusted for a range of factors to understand further inequalities in vaccination coverage.
- The main limitation is that most demographic and socio-economic characteristics were derived from the 2011 Census and therefore are ten years old
- Because the dataset is based on the 2011 Census, it excluded people living in England in 2011 but not taking part in the 2011 Census, respondents who could not be linked to the 2011-2013 NHS patients register and recent migrants.